# One-step multiplex PCR assay for identification of *Mycobacterium kansasii* complex species

Paulina Joanna Wałpuska,[1] Zofia Bakuła,[1] Katarzyna Rastawicka,[1] Tomasz Jagielski[1]

**ABSTRACT** Currently, the *Mycobacterium kansasii* complex (MKC) species identification involves PCR amplification, followed by sequencing or digestion of the amplicons. The purpose of this study was to develop a one-step multiplex PCR assay allowing for a fast and accurate identification of MKC species. A total of 158 *Mycobacterium* sp. genomes were searched for sites that would yield easily detectable amplicons of different sizes among the MKC species while producing no amplicons for other-than-MKC mycobacterial species. Three primer sets were designed and tested *in vitro* on seven reference strains, representing all MKC species. For the evaluation of the method, a total of 136 *Mycobacterium* sp. isolates were analyzed. The vast majority (96/98, 98%) of the tested MKC strains yielded a species-characteristic profile, consistent with *in silico* predictions. The only two isolates, which gave incongruent results, belong to atypical *M. kansasii* subtype IIB and were designated as *M. persicum* with a newly designed assay. No products were obtained for mycobacteria other than MKC. This study offers a new PCR-based method for identification of all MKC species. It involves a single-step protocol and yields reproducible and easily interpretable results.

**IMPORTANCE** Current methods used for the *Mycobacterium kansasii* complex (MKC) species identification lack the resolution to differentiate between individual MKC species and require time-demanding multiple steps and costly analyses. Herein, we present a novel one-step multiplex PCR assay that enables rapid and cost-effective species-level identification of MKC members, suitable for implementation in both clinical diagnostics and epidemiological investigations. The method was evaluated using 143 *Mycobacterium* strains, including 17 reference strains and 126 clinical or environmental isolates. It accurately identified the vast majority (98%) of strains, with only two atypical *M. kansasii* subtype IIB isolates misidentified as *M. persicum*. The calculated sensitivity and specificity of the assay were 98% and 100%, respectively.

**KEYWORDS** *Mycobacterium kansasii* complex (MKC), species identification, multiplex PCR (mPCR), non-tuberculous mycobacteria (NTM)

The group of non-tuberculous mycobacteria (NTM) consists of over 200 *Mycobacterium* species that do not cause tuberculosis or leprosy (1). Although NTM are for the most part free-living saprophytes widespread in nature, they are particularly well adapted to human-engineered environments, such as drinking water distribution systems and plumbing installations in residential, commercial, and public buildings (2, 3). NTM infections or mycobacterioses pose an increasing global health concern, and the bacteria themselves are considered as emerging pathogens (4). Accurate diagnosis of NTM disease relies upon an in-depth medical and laboratory investigation, which integrates clinical, radiological, and microbiological evidence (5, 6). Methods allowing for a fast and reliable NTM species identification are still being sought.

**Peer Reviewers** Harley T. Harris, University of Utah Health, Salt Lake City, Utah, USA; La'Tonzia L. Adams, Johns Hopkins University School of Medicine, Baltimore, Maryland, USA

Address correspondence to Tomasz Jagielski, t.jagielski@uw.edu.pl, or Zofia Bakuła, zofiabakula@uw.edu.pl.

The authors declare no conflict of interest.

See the funding table on p. 7.

One of the most frequent NTM pathogens isolated from clinical samples throughout the world is *Mycobacterium kansasii* (7). The species, originally described in the 1950s, was later found to accommodate six distinct genotypes or subtypes (I–VI), which recently have been elevated to species rank, based on whole-genome sequence analysis (8). Consequently, the *Mycobacterium kansasii* complex (MKC) has been established, encompassing six former *M. kansasii* subtypes, namely, *M. kansasii* (former type I), *M. persicum* (II), *M. pseudokansasii* (III), *M. ostraviense* (IV), *M. innocens* (V), and *M. attenuatum* (VI). The MKC has also been expanded to include *M. gastri* (8).

Currently, there are two major molecular-based methods used for MKC identification. One is PCR sequencing of either *tuf*, *hsp65*, or *rpoB* genes, coding for the thermo-unstable elongation factor, the 65 kDa heat shock protein, or the β-subunit of RNA polymerase, respectively, or the 16S-23S rDNA internal transcribed spacer within the rDNA locus (9–11). The other is PCR-restriction enzyme analysis (PCR-REA), which involves amplification of partial *tuf*, *hsp65*, or *rpoB* genes, followed by digestion of their amplicons with a combination of one, two, or three restriction enzymes (MvaI for *tuf*, BstEII and HaeIII for *hsp65*, and MvaI, AccII, and HaeIII for *rpoB*) (10–13). The recently proposed one-step multiplex PCR (mPCR) assay includes only five MKC species (all except *M. attenuatum* and *M. gastri*) (14). The assay was not tested on MKC reference strains, did not include other-than-MKC NTM strains, and was evaluated on a very small pool of clinical strains (15 in total). These limitations indicate that the method has not been comprehensively validated.

The purpose of this work was twofold: (i) to validate the previously developed mPCR assay using MKC reference strains and (ii) to design a new alternative to PCR-REA and PCR-sequencing, mPCR assay, allowing for accurate identification of each of the MKC species.

## MATERIALS AND METHODS

### Evaluation of a previously designed mPCR assay

#### Isolates, culturing, and DNA isolation

Two sets of isolates were used for the validation of the previously designed method (14). One set included the reference strains of five MKC species, i.e., *M. kansasii* (ATCC 12478T), *M. persicum* (B11063838), *M. pseudokansasii* (MK142), *M. ostraviense* (241/15), and *M. innocens* (MK13) (Table S1). The second set included five MKC Kaust (I–V) strains, kindly provided by the authors of the method (14). All strains were cultured on Löwenstein-Jensen agar medium with standard mycobacteriological procedures. Genomic DNA was extracted using a modified cetyl-trimethyl-ammonium bromide method, as described elsewhere (12). The purified DNA was dissolved in the TE buffer and quantified with the NanoDrop 2000 Spectrophotometer (Thermo Fisher Scientific, USA). The DNA samples were diluted (*ca*. 10 ng/µL) and stored at −20°C until use.

#### Method validation

In accordance with the previously designed mPCR assay, a set of six primer pairs was used to generate species-specific amplicon patterns for five MKC species (14). The reactions were performed with an OptiTaq Master Mix kit (EURx, Poland), as recommended by the manufacturer, in 30 µL reaction mixtures containing *ca*. 10 ng of bacterial DNA. The DNA fragments were electrophoresed on 2% agarose-Tris-borate-EDTA gels at 120 V for 50 min, then visualized by staining with ethidium bromide and UV fluorescence. Band sizes were determined manually by comparing DNA migration to a molecular marker.

#### Species identification

For species confirmation of the Kaust I–V strains, PCR-REA of the *tuf* gene was applied as described previously (12). In brief, partial (740 bp) *tuf* gene amplicons were obtained

with Color Taq PCR Master Mix (2×) (EURx) in 30 µL reaction mixtures, consisting of *ca*. 10 ng of bacterial DNA and a T1/T2 primer pair. The products were then digested with Fast Digest MvaI enzyme (Thermo Scientific, USA). The DNA fragments were electrophoresed on 4% agarose-Tris-acetate-EDTA gels at 120 V for 90 min and visualized as in the Method Validation section.

## Design of a new assay

### *In silico analysis*

The study included 158 *Mycobacterium* sp. genomes deposited in the GenBank database (https://www.ncbi.nlm.nih.gov/genbank/), as listed in Table S2. First, genomes of seven reference isolates (marked as "T" in Table S2) were screened using a custom-designed script to identify sites producing easily detectable amplicons (100–1,000 bp) that differed in size among MKC species (minimum difference of 30 nucleotides between products) while yielding no amplicons in non-MKC mycobacterial species. Sites containing stretches of five consecutive G + C or A + T nucleotides were excluded. Primers were designed to amplify the selected sites and met the following criteria: (i) length of 19–28 nucleotides; (ii) melting temperature of 50°C–65°C, with a maximum difference of 2°C between primer pairs; (iii) GC content of 40%–60%; and (iv) the terminal three nucleotides at both the 5′ and 3′ ends of each primer were not exclusively G + C. The designed primer pairs were then screened against all 158 *Mycobacterium* genomes (Table S2) using Bowtie 2 software (15), resulting in the final primer set (Table S3).

### *Isolates, culturing, and DNA isolation*

Three designed primer pair sets were tested *in vitro* on seven reference strains, representing all MKC species, to select the most efficient pair. The method was further validated on 136 *Mycobacterium* spp. isolates (Table S1), including reference strains of 8 non-MKC NTM and 2 *M. tuberculosis* complex (MTBC) species. Overall, 98 MKC, 35 NTM other-than-MKC, and 3 MTBC strains were included in the analysis, representing 6, 21, and 2 species, respectively. All strains had been identified to the species level either via whole-genome sequencing (WGS), or *tuf*, *hsp65*, or *rpoB* PCR-sequencing or PCR-REA analysis (8) (12). Culturing and DNA extraction were carried out as described in the analogous section Evaluation of the Previously Designed mPCR Assay.

### *Method validation*

The PCR reactions were set up with a Color Taq PCR Master Mix (2x) (EURx), as recommended by the manufacturer, supplemented with 1M final concentration of betaine (Merck, Germany), in 30 µL reaction mixtures containing *ca*. 10 ng of bacterial DNA. The cycling conditions included preheating at 95°C for 3 min, followed by 30 cycles of denaturation at 94°C for 30 s, annealing at 54°C for 30 s, and elongation at 72°C for 45 s. The final elongation step was performed at 72°C for 7 min. The DNA fragments were electrophoresed and visualized as described in the analogous section Evaluation of the Previously Designed mPCR Assay.

## RESULTS

## Evaluation of the previously designed mPCR assay

Out of five MKC reference strains, only *M. kansasii* and *M. ostraviense* yielded species-specific PCR amplicons, yet accompanied by other non-specific products. For other MKC reference strains (i.e., *M. persicum*, *M. pseudokansasii*, and *M. innocens*), the amplicon sizes differed from those predicted *in silico* and obtained *in vitro* by Guan et al. (Fig. 1).

To verify the accuracy of the method's design, Kaust I–V strains (originally identified by Guan et al. as *M. kansasii*, *M. persicum*, *M. pseudokansasii*, *M. ostraviense*, and *M. innocens*, respectively) were identified using *tuf*-based PCR-REA. The species identity was confirmed only for strains Kaust I and IV. The remaining three Kaust strains were

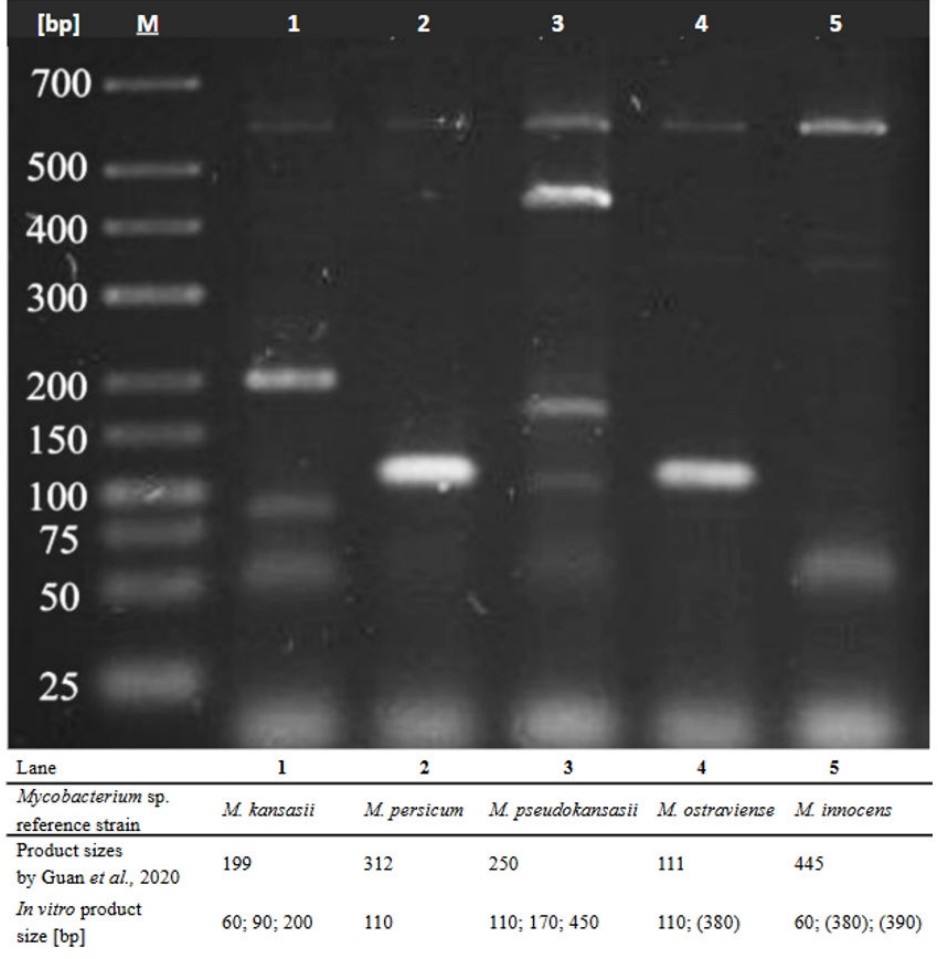

| Lane | 1 | 2 | 3 | 4 | 5 |
|---|---|---|---|---|---|
| *Mycobacterium* sp. reference strain | *M. kansasii* | *M. persicum* | *M. pseudokansasii* | *M. ostraviense* | *M. innocens* |
| Product sizes by Guan *et al.*, 2020 | 199 | 312 | 250 | 111 | 445 |
| *In vitro* product size [bp] | 60; 90; 200 | 110 | 110; 170; 450 | 110; (380) | 60; (380); (390) |

**FIG 1** Evaluation of the previously designed one-step multiplex PCR assay by Guan et al. (14) on *M. kansasii* complex reference strains. Lanes 1–5, MKC reference strains; 590 bp product, inner reaction control using primers specific to the 16S rRNA gene; <u>M</u>, GeneRuler 100 bp DNA Ladder.

identified as follows: Kaust II, *M. kansasii*; Kaust III, *M. avium*; and Kaust V, *M. pseudokansasii*.

## Design of a new assay

### In silico predictions and evaluation on MKC reference strains

Upon *in silico* simulations, three sets of primer pairs were designed (Table S3). All these primer sets were tested *in vitro* using seven reference strains, representing all MKC species (Table S1). Despite multiple attempts to optimize the PCR conditions, primer sets nos. 1 and 3 produced PCR products inconsistent with *in silico* predictions. Therefore, they were not considered for further investigation. For reactions with primers set no. 2, which gave results most congruent with *in silico* simulations, it was chosen for further investigation. The reaction parameters and cycling conditions with the chosen primers were optimized to ensure efficient amplification of products of the desired size while minimizing non-specific amplification (Fig. 2).

### In vivo validation

To evaluate the newly designed method, the assay with primer pair no. 2 was further tested on a set of 136 *Mycobacterium* sp. isolates (Table S1). The vast majority of the analyzed MKC isolates (96/98, 98%) were correctly assigned to the species level. The

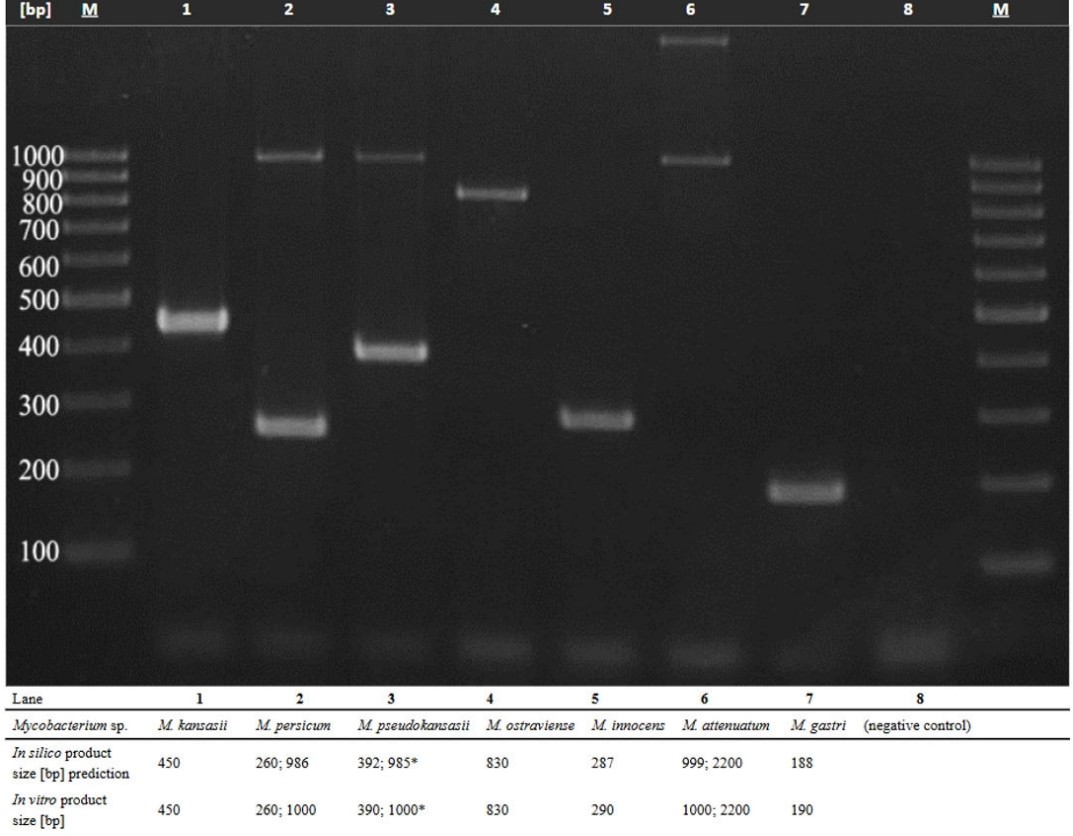

| Lane | 1 | 2 | 3 | 4 | 5 | 6 | 7 | 8 |
|---|---|---|---|---|---|---|---|---|
| *Mycobacterium* sp. | *M. kansasii* | *M. persicum* | *M. pseudokansasii* | *M. ostraviense* | *M. innocens* | *M. attenuatum* | *M. gastri* | (negative control) |
| *In silico* product size [bp] prediction | 450 | 260; 986 | 392; 985* | 830 | 287 | 999; 2200 | 188 | |
| *In vitro* product size [bp] | 450 | 260; 1000 | 390; 1000* | 830 | 290 | 1000; 2200 | 190 | |

**FIG 2** A newly designed one-step multiplex PCR (mPCR) assay profiling of representatives of *M. kansasii* complex species. Lanes 1–7, MKC reference strains; lane 8, negative control (no DNA); *, an occasionally occurring *ca.* 1,000 bp band for mPCR of *M. pseudokansasii* species; lane 8, negative control (no DNA); M, GeneRuler 100 bp DNA Ladder.

only two isolates which gave incongruent results (*in silico* vs *in vitro*) were *M. kansasii* subtype IIB strains K14 and K19 (listed in Table S1 as 53 and 54, respectively), which were incorrectly identified as *M. persicum* with a newly designed mPCR (Fig. S1 and S2). For the NTM isolates outside the MKC group, the new mPCR assay generated no bands (Fig. S3). Likewise, no amplification products were detected for two MTBC species (Fig. S3).

The calculated sensitivity and specificity of the assay were 98% and 100%, respectively.

## DISCUSSION

The taxonomy of MKC has recently been revised, reclassifying the former six subtypes (I–VI) as six distinct species (8). This change reflects differences in the pathogenic potential of the MKC species. *M. kansasii* and *M. persicum* are the most frequently associated with human disease, while the remaining species, i.e., *M. pseudokansasii*, *M. ostraviense*, *M. innocens*, and *M. attenuatum*, are primarily environmental or considered contaminants, rarely implicated in human infections. Thus, accurate species-level identification within the MKC has become essential for guiding clinical decision-making and ensuring meaningful epidemiological surveillance (7, 16).

Among the methods currently used for the identification of MTBC and NTM species in clinical settings, line probe assays, such as INNO LiPA MYCOBACTERIA v2 (Innogenetics, Ghent, Belgium) (17), Speed-oligo Mycobacteria (Vircell, Santa Fé Granada, Spain) (18), and GenoType *Mycobacterium* CM/AS (Hain Lifescience, Nehren, Germany) (19), have been widely used due to high accuracy and relatively rapid turnaround time. Despite the overall diagnostic value of these assays, they lack the resolution to differentiate

between individual species within the MKC. Thus, PCR-sequencing and PCR-REA are still the only two easy-to-use, molecular methods allowing for identification of MKC species (20–22). These techniques, however, are multi-step procedures and therefore time- and resource-consuming (23, 24).

In response to these limitations, the herein proposed one-step mPCR emerges as a promising alternative offering faster and more cost-effective identification of MKC species. Over the last decade, several mPCR-based methods have been developed for identification of clinically relevant mycobacteria (25–30). For instance, Kim et al. (28) designed an assay for the detection of *Mycobacterium* sp. and further differentiation of *M. tuberculosis*, *M. bovis*, and *M. africanum* (28). Another mPCR was developed to detect *M. tuberculosis* and species within *M. avium* complex in the first reaction, and *M. kansasii*, *M. fortuitum*, *M. abscessus*, and *M. massiliense* in the second reaction (23). Two other studies focused on separating *M. tuberculosis* from NTM collectively (29, 30).

The only one-step mPCR assay designed for MKC species identification was developed by Guan et al. (14). It allowed differentiation between five out of seven MKC species (i.e., *M. kansasii*, *M. persicum*, *M. pseudokansasii*, *M. ostraviense*, and *M. innocens*) and was tested *in vitro* on 5 environmental (Kaust I–V) and 15 clinical strains only. Importantly, the method has never been tested on reference strains and did not include any strains outside MKC. The present study is the first to validate the assay using the reference MKC strains. Only *M. kansasii* and *M. ostraviense* produced species-specific PCR amplicons, though with additional non-specific products. Applying PCR-REA-based identification, we found that three out of five Kaust strains (Kaust II, III, and V) were shown to represent other species than expected. These findings cast doubt on the accuracy and reliability of the method (14).

Here, a novel one-step mPCR assay was developed to enable rapid and unambiguous species identification of all members of MKC. The assay accurately identified 96 out of 98 MKC strains, corresponding to a specificity of 98%. Moreover, none of the non-MKC isolates, including those from the MTBC, were misidentified as MKC, indicating a sensitivity of 100%.

Strains that produced non-specific results (K14 and K19) had previously been dubbed atypical *M. kansasii* subtype IIB. These strains yielded discordant identification results. Upon PCR-REA of *tuf* and *rpoB* genes, profiles characteristic of *M. persicum* were observed (13), whereas WGS analysis classified the two strains as *M. kansasii* (8). Interestingly, two other strains described in the literature produced conflicting identification results and were classified as atypical type I (Ib) and intermediate type I (I/II) (8, 13). These two strains were correctly identified as *M. kansasii* with our mPCR assay.

For some of the *M. pseudokansasii* strains (Fig. S2), including the reference strain (Fig. 2), a *ca*. 1,000 bp band was detected, congruently with the *in silico* analyses. However, this band was occasionally absent or exhibited very low intensity. Nevertheless, unsystematic lack of this band does not affect the interpretation of the assay, as the fragment of *ca*. 390 bp constitutes the relevant discriminating product for *M. pseudokansasii*.

The proposed method has several limitations. First, the number of strains representing less common MKC members (i.e., *M. ostraviense*, *M. innocens*, and *M. gastri*), used for the evaluation of the assay, was low. Likewise, the number of species/isolates of NTM other than MKC and MTBC was not exhaustive. Second, the mPCR assays are generally highly vulnerable to PCR conditions. Therefore, some non-specific products may occasionally occur, which can interfere with the accuracy of the results. Third, the absence of a PCR product may be misinterpreted as a non-MKC due to a technical issue. To minimize this risk, we recommend strictly following the experimental conditions and running a separate positive control (e.g., using DNA from an MKC reference strain and *hsp65*-based PCR [31]) irrespective of the band presence upon analysis.

In summary, the proposed mPCR assay serves as a rapid, cost-effective species identification tool. It can be applied following the initial MKC detection in the analyzed sample (clinical or environmental) by phenotypic methods, commercial line probe

assays, or MALDI-TOF MS. Its one-step design and minimal equipment requirements make it suitable for routine diagnostic laboratories as well as research settings. Contrary to existing multi-step methods, it provides fast (with a turnaround time of less than 3 h vs 1–3 days), affordable (with a per-reaction cost of *ca.* 2 EUR vs 5–60 EUR), and reliable (98% specificity vs 75%–100%, depending on the applied method, target gene, and reference databases) identification of all MKC species (32–34). This method might be implemented in laboratories with basic molecular biology infrastructure (i.e. thermocycler, electrophoresis, and gel imaging systems) as an alternative to PCR-REA and PCR-sequencing.

## ACKNOWLEDGMENTS

The authors thank Dr. Guan for kindly providing the Kaust strains.

The study was supported by two grants under the "Initiative of Excellence – Research University" program (nos. BOB-IDUB-622-616/2023 and BOB-IDUB-622-913/2025) and by the "SONATA" grant from the National Science Center (no. 2021/43/D/NZ6/01250). The funders had no role in study design, data collection, and interpretation.

## AUTHOR AFFILIATION

[1]Department of Medical Microbiology, Institute of Microbiology, Faculty of Biology, University of Warsaw, Warsaw, Poland

## AUTHOR ORCIDs

Paulina Joanna Wałpuska  http://orcid.org/0000-0001-6165-8330
Zofia Bakuła  http://orcid.org/0000-0001-7560-2100
Tomasz Jagielski  http://orcid.org/0000-0001-9553-5742

## FUNDING

| Funder | Grant(s) | Author(s) |
|---|---|---|
| National Science Centre (NCN) "SONATA" | 2021/43/D/NZ6/01250 | Zofia Bakuła |
| "The Excellence Initiative - Research University" (IDUB) programme | BOB-IDUB-622-616/2023 | Paulina Joanna Wałpuska |

## ADDITIONAL FILES

The following material is available online.

### Supplemental Material

**Supplemental material (Spectrum03267-25-s0001.pdf).** Tables S1 to S3; Fig. S1 to S3.

### Open Peer Review

**PEER REVIEW HISTORY (review-history.pdf).** An accounting of the reviewer comments and feedback.

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
