## [Reviewer comments · Microbiology Spectrum]

Microbiology Spectrum

One-step multiplex PCR assay for identification of *Mycobacterium kansasii* complex species

Paulina Walpuska, Zofia Bakuła, Katarzyna Rastawicka, and Tomasz Jagielski

Corresponding Author(s): Tomasz Jagielski, Institute of Microbiology, Faculty of Biology, University of Warsaw

Review Timeline:

Submission Date:	October 9, 2025
Editorial Decision:	December 1, 2025
Revision Received:	December 23, 2025
Accepted:	February 6, 2026

Editor: Natalie Whitfield

Reviewer(s): Disclosure of reviewer identity is with reference to reviewer comments included in decision letter(s). The following individuals involved in review of your submission have agreed to reveal their identity: Harley T Harris (Reviewer #1); La'Tonzia L Adams (Reviewer #2)

Transaction Report:

DOI: <https://doi.org/10.1128/spectrum.03267-25>

Re: Spectrum03267-25 (One-step multiplex PCR assay for identification of *Mycobacterium kansasii* complex species)

Dear Dr. Tomasz Jagielski:

Thank you for the privilege of reviewing your work. Below you will find my comments, instructions from the Spectrum editorial office, and the reviewer comments.

When submitting the revised version of your paper, please provide point-by-point responses addressing the issues raised by the reviewers.

Revision Guidelines

Sincerely,
Natalie Whitfield
Editor
Microbiology Spectrum

Reviewer #1 (Comments for the Author):

Major Comments for Authors:

- The utilization/ importance of this multiplex PCR is not clearly mentioned until the discussion section (line 151-152). This should be included in the Importance section and introduction to explain why this assay is needed. Also, authors should expand more

when this assay should be utilized. Is this for use in a diagnostic setting or research area? Is this to be used in conjunction with other identification methods such as after a MALDI or phenotypic methods? If for clinical usage, authors should also address the utility of this method such as access to ethidium bromide and fluorescent gel imagers.

- Line 145-147 and Supplemental Fig 1: Authors note no amplification for non-MKC NTMs and MTBC however this data is not included in supplemental figures. Please add in the appropriate figures, ideally as a supplemental figure 2 since the first figure is already lengthy.
- Methods section has some areas needing further information. For band size measurement (line 83-84), were band sizes determined by manual measurement or was software utilized for calculation? In describing the primer pair testing (line 102-103 and 134-135), what criteria were used to evaluate the effectiveness of the primer pairs i.e. No off-target amplification, band size consistent with prediction, no overlap of sizes between species?
- Line 202-203: Authors note that utilizing a positive control such as a separate target PCR to ensure PCR was not inhibited. However, it appears the authors did not use a separate target control outside of figure 1 using the previously published method. As this control is needed for separating failed PCR from non-MKC NTM or MTBC, authors should show at least for the reference strains tested in fig 2 the utilization of an internal control and how it impacts the reading of other bands.
- Supplemental Figure 1 and its discussion need to be reworked for clearer understanding of readers.
 - o The figure legend needs to be further detail of what is shown, indicating it is the new method and testing against reference and/or clinical strains.
 - o Add in subfigure letters such as panel A, B, C, etc. so readers can easily determine what figures remarks go with and reduce confusion. Notably, I am uncertain if the alternative ladder is only for the last *M. kansasii* figure panel or if it applies to the *M. persicum* after it.
 - o The subheadings for each species should be above the figures associated with or included in the legend and then referenced according to panel letter. Also including the expected band size would keep the reader from needing to flip back for reference to figure 2. For example, "panels A-E are *M. kansasii* (expected band size- 450 bp)".
 - o Explanations of the X's in second figure panel are needed. Were these failed attempts for those shown in next panel? If so, should mention if this resolved on repeat or if multiple attempts were needed.
 - o Most importantly, there is no discussion or explanation of why the ~1000 bp band is missing from almost all the *M. psuedokansasii*. This needs to be addressed in the text as while the second band is different in size from any others, amplification occurred with the reference strain and in other species longer amplicons were generated making it unlikely to be length related. This needs to be discussed so if someone was implementing the assay and only got one band they would know if this is or is not a valid *M. psuedokansasii* determination.

Minor Comments for Authors:

- Line 12: expand to *Mycobacterium kansasii* as first presentation.
- Line 91, 108, 115: this section notation is not used in the paper formatting which creates confusion. Using "as described above" or "as described in ____ section".
- Line 104: MTBC needs to be expanded as first introduction of the abbreviation.
- Line 134: Add reference to Supplemental Table 1 for the reference strains used.
- Line 144: add notation that K14 and K19 are isolates number 53 and 54 respectively.
- Figure 1: make the 590 bp product notation match between figure and text. Either state as "boxed bands" or remove the box on figure and replace with *.
- Figure 2: What is the * for in *M. psuedokansasii* column?

Reviewer #2 (Comments for the Author):

1. Novelty and Contribution

The development of a one-step multiplex PCR assay is a meaningful advancement for MKC diagnostics. Consider emphasizing how the assay compares (cost, turnaround, complexity) with existing commercial or laboratory-developed tests.

2. Handling of Atypical Strains

The misidentification of the two subtype IIB isolates is important. The abstract mentions these were reassigned as *M. persicum* using a newly designed assay-however, this could confuse readers. A brief clarification that this newer assay is ancillary and not part of the main single-step method might prevent misinterpretation.

Journal Name: Microbiology Spectrum

Manuscript Number: Spectrum03267-25

Manuscript Title: One-step multiplex PCR assay for identification of *Mycobacterium kansasii* complex species

Overall Impressions and comments to editor:

This paper has two focuses: evaluating a previously published PCR method for *M. kansasii* complex species identification and presenting and evaluating a novel PCR method to include all 7 species. Similar studies have been done before for *M. tuberculosis* complex and non-tuberculous *Mycobacteria*, including the methodology tested in the first part of the paper against a reference set of isolates. The novel method presented by the authors builds on the prior test method to improve the methodology and includes *M. attenuatum* and *M. gastri*. Overall, this has potential to be an impactful assay to identify the species of *M. kansasii* complex. However, the authors need to address some drawbacks of the paper. The supplemental figure is missing data that should be included according to the text and is poorly organized for full understanding by the reader as it is unclear where some comments refer to. Authors also need to add some further information to the methods section and utilization of the test. With this, the paper will need modifications prior to a final decision being made on the acceptance of the paper.

Major Comments for Authors:

- The utilization/ importance of this multiplex PCR is not clearly mentioned until the discussion section (line 151-152). This should be included in the Importance section and introduction to explain why this assay is needed. Also, authors should expand more when this assay should be utilized. Is this for use in a diagnostic setting or research area? Is this to be used in conjunction with other identification methods such as after a MALDI or phenotypic methods? If for clinical usage, authors should also address the utility of this method such as access to ethidium bromide and fluorescent gel imagers.
- Line 145-147 and Supplemental Fig 1: Authors note no amplification for non-MKC NTMs and MTBC however this data is not included in supplemental figures. Please add in the appropriate figures, ideally as a supplemental figure 2 since the first figure is already lengthy.
- Methods section has some areas needing further information. For band size measurement (line 83-84), were band sizes determined by manual measurement or was software utilized for calculation? In describing the primer pair testing (line 102-103 and 134-135), what criteria were used to evaluate the effectiveness of

the primer pairs i.e. No off-target amplification, band size consistent with prediction, no overlap of sizes between species?

- Line 202-203: Authors note that utilizing a positive control such as a separate target PCR to ensure PCR was not inhibited. However, it appears the authors did not use a separate target control outside of figure 1 using the previously published method. As this control is needed for separating failed PCR from non-MKC NTM or MTBC, authors should show at least for the reference strains tested in fig 2 the utilization of an internal control and how it impacts the reading of other bands.
- Supplemental Figure 1 and its discussion need to be reworked for clearer understanding of readers.
 - The figure legend needs to be further detail of what is shown, indicating it is the new method and testing against reference and/or clinical strains.
 - Add in subfigure letters such as panel A, B, C, etc. so readers can easily determine what figures remarks go with and reduce confusion. Notably, I am uncertain if the alternative ladder is only for the last *M. kansasii* figure panel or if it applies to the *M. persicum* after it.
 - The subheadings for each species should be above the figures associated with or included in the legend and then referenced according to panel letter. Also including the expected band size would keep the reader from needing to flip back for reference to figure 2. For example, “panels A-E are *M. kansasii* (expected band size- 450 bp)”.
 - Explanations of the X’s in second figure panel are needed. Were these failed attempts for those shown in next panel? If so, should mention if this resolved on repeat or if multiple attempts were needed.
 - Most importantly, there is no discussion or explanation of why the ~1000 bp band is missing from almost all the *M. psuedokansasii*. This needs to be addressed in the text as while the second band is different in size from any others, amplification occurred with the reference strain and in other species longer amplicons were generated making it unlikely to be length related. This needs to be discussed so if someone was implementing the assay and only got one band they would know if this is or is not a valid *M. psuedokansasii* determination.

Minor Comments for Authors:

- Line 12: expand to *Mycobacterium kansasii* as first presentation.
- Line 91, 108, 115: this section notation is not used in the paper formatting which creates confusion. Using “as described above” or “as described in ____ section”.
- Line 104: MTBC needs to be expanded as first introduction of the abbreviation.
- Line 134: Add reference to Supplemental Table 1 for the reference strains used.

- Line 144: add notation that K14 and K19 are isolates number 53 and 54 respectively.
- Figure 1: make the 590 bp product notation match between figure and text. Either state as “boxed bands” or remove the box on figure and replace with *.
- Figure 2: What is the * for in *M. psuedokansasii* column?

21st December 2025

Dr. Natalie N. Whitfield

Editor

Microbiology Spectrum

Dear Dr. Whitfield,

On behalf of the authors I would like to thank the Reviewers for thorough evaluation of our manuscript entitled *One-step multiplex PCR assay for identification of Mycobacterium kansasii complex species* (#Spectrum03267-25). You will find below our answers (highlighted in grey) to the Reviewers' comments. We hope our revisions adequately address the Reviewers' concerns.

Comments from the Reviewer #1

Major revisions

- The utilization/importance of this multiplex PCR is not clearly mentioned until the discussion section (**line 151-152**). This should be included in the Importance section and introduction to explain why this assay is needed. Also, authors should expand more when this assay should be utilized. Is this for use in a diagnostic setting or research area? Is this to be used in conjunction with other identification methods such as after a MALDI or phenotypic methods? If for clinical usage, authors should also address the utility of this method such as access to ethidium bromide and fluorescent gel imagers.

Following the Reviewer's remarks, we have now modified the "**Importance**", "**Introduction**", and "**Discussion**" sections and added 3 positions to the "**References**", accordingly:

Importance (lines 28-32):

"Current methods used for the M. kansasii complex (MKC) species identification lack the resolution to differentiate between individual MKC species, require time-demanding multiple steps and costly analyses. Herein, we present a novel one-step mPCR assay that enables rapid and cost-effective species-level identification of MKC members, suitable for implementation in both clinical diagnostics and epidemiological investigations."

Introduction (lines 66-68):

"The purpose of this work was two-fold: (i) to validate the previously developed mPCR assay using MKC reference strains and (ii) to design a new, alternative to PCR-REA and PCR-sequencing, mPCR assay, allowing for accurate identification of each of the MKC species."

Discussion (lines 229-238):

“In summary, the proposed mPCR assay serves as a rapid, cost-effective species identification tool. It can be applied following the initial MKC detection in the analyzed sample (clinical or environmental) by phenotypic methods, commercial line probe assays, or MALDI-TOF MS. Its one-step design and minimal equipment requirements make it suitable for routine diagnostic laboratories as well as research settings. Contrary to existing multi-step methods, it provides fast (with a turnaround time of less than 3 hours vs. 1-3 days), affordable (with a per-reaction cost of ca. 2 EUR vs. 5-60 EUR), and reliable (98% specificity vs. 75-100%, depending on the applied method, target gene, and reference databases) identification of all MKC species (32-34). This method might be implemented in laboratories with basic molecular biology infrastructure (i.e. thermocycler, electrophoresis and gel imaging systems) as an alternative to PCR-REA and PCR-sequencing.”

References:

32. Alcolea-Medina A, Fernandez MTC, Montiel N, García MPL, Sevilla CD, North N, Lirola MJM, Wilks M. An improved simple method for the identification of Mycobacteria by MALDI-TOF MS (Matrix-Assisted Laser Desorption- Ionization mass spectrometry). *Sci Rep.* 2019. 9(1):20216. doi:10.1038/s41598-019-56604-7

33. Li B, Zhu C, Sun L, Dong H, Sun Y, Cao S, Zhen L, Qi Q, Zhang Q, Mo T, Wang H, Qiu M, Song C, Cai Q. Performance evaluation and clinical validation of optimized nucleotide MALDI-TOF-MS for mycobacterial identification. *Front Cell Infect Microbiol.* 2022. 12:1079184. doi: 10.3389/fcimb.2022.1079184.

34. Liu X, Niu H, Guo D, Gao H, Wu L, Liu J, Bai C, Li Y, Wang P, Zhou Z, Wang Y, Liang J, Gong W. Application value of nucleic acid MALDI-TOF MS in mycobacterial species identification and drug resistance detection in *Mycobacterium tuberculosis*. *Microbiol Spectr.* 2025. 13(5):e0154524. doi: 10.1128/spectrum.01545-24

- **Line 145-147 and Supplemental Figure 1:** Authors note no amplification for non-MKC NTMs and MTBC however this data is not included in supplemental figures. Please add in the appropriate figures, ideally as a **Supplemental Figure 2** since the first figure is already lengthy.

As recommended by the Reviewer, we have divided **Suppl. Fig. 1** into **Suppl. Figs. 1-3** to present the electrophoretic gels for each mycobacterial subgroup separately. **Suppl. Figs. 1-3** now display gels for: (i) *M. kansasii*, (ii) other MKC members (excluding *M. kansasii*), and (iii) non-MKC NTMs and MTBC, respectively.

- Methods section has some areas needing further information. For band size measurement (**line 83-84**), were band sizes determined by manual measurement or was software utilized for calculation? In describing the primer pair testing (**line 102-103 and 134-135**), what

criteria were used to evaluate the effectiveness of the primer pairs i.e. no off-target amplification, band size consistent with prediction, no overlap of sizes between species?

We have now included the information on band size measurement in the “**Methods**” sections “**Evaluation of a previously designed mPCR assay: Method validation**”. Please, see **lines 88-89**:

“Band sizes were determined manually, by comparing DNA migration to a molecular marker.”

Furthermore, we have now provided an extended explanation on the primers design criteria in the “**Design of a new assay: In silico analysis**” lines **99-110**:

*“The study included 158 Mycobacterium sp. genomes deposited in the GenBank database (<https://www.ncbi.nlm.nih.gov/genbank/>), as listed in **Suppl. Tab. 2**. First, genomes of 7 reference isolates (marked as “T” in the **Suppl. Tab. 2**.) were screened using a custom-designed script to identify sites producing easily detectable amplicons (100–1000 bp) that differed in size among MKC species (minimum difference of 30 nucleotides between products), while yielding no amplicons in non-MKC mycobacterial species. Sites containing stretches of five consecutive G+C or A+T nucleotides were excluded. Primers were designed to amplify the selected sites and met the following criteria: (i) length of 19–28 nucleotides; (ii) melting temperature (T_m) of 50–65 °C, with a maximum difference of 2 °C between primer pairs; (iii) GC content of 40–60%; and (iv) the terminal three nucleotides at both the 5' and 3' ends of each primer were not exclusively G+C. The designed primer pairs were then screened against all 158 Mycobacterium genomes (**Suppl. Tab. 2**) using Bowtie 2 software (15), resulting in the final primer set (**Suppl. Tab. 3**).*

We have also clarified the evaluation of the designed primers in the “**Results**” section (**lines 146-150**):

“Despite multiple attempts to optimize the PCR conditions, primer sets no. 1 and 3 produced bands inconsistent with in silico predictions. Therefore, they were not considered for further investigation.”

□ **Line 202-203**: Authors note that utilizing a positive control such as a separate target PCR to ensure PCR was not inhibited. However, it appears the authors did not use a separate target control outside of figure 1 using the previously published method. As this control is needed for separating failed PCR from non-MKC NTM or MTBC, authors should show at least for the reference strains tested in fig 2 the utilization of an internal control and how it impacts the reading of other bands.

The multiplex PCR assay designed here was not developed with an internal positive control. However, following the Reviewer’s suggestion, we have now performed an additional analysis for the reference strains, using primer pair no. 2 along with primers for a fragment of *hsp65* (31). Since we have obtained unspecific PCR products, such a mix cannot be used for MKC species identification.

Therefore, if no PCR band is observed in the analysis, we recommend running a separate PCR with a positive control. This now has been addressed in the “**Discussion**” section (**lines 224-227**):

“Thirdly, the absence of a PCR product may be misinterpreted as a non-MKC due to a technical issue. To minimize this risk, we recommend strictly following the experimental conditions

and running a separate positive controls (e.g., using DNA from an MKC reference strain and hsp65-based PCR (31)) irrespective of the band presence upon analysis.”

Accordingly, a reference no. 31 was added to the “References” section:

31. Kim H, Kim SH, Shim TS, Kim MN, Bai GH, Park YG, Lee SH, Chae GT, Cha CY, Kook YH, Kim BJ. Differentiation of *Mycobacterium* species by analysis of the heat-shock protein 65 gene (*hsp65*). *Int J Syst Evol Microbiol*. 2005;55(Pt 4):1649-1656. doi:10.1099/ijs.0.63553-0.

- **Supplemental Figure 1** and its discussion need to be reworked for clearer understanding of readers: The figure legend needs to be further detail of what is shown, indicating it is the new method and testing against reference and/or clinical strains.

Add in subfigure letters such as panel A, B, C, etc. so readers can easily determine what figures remarks go with and reduce confusion. Notably, I am uncertain if the alternative ladder is only for the last *M. kansasii* figure panel or if it applies to the *M. persicum* after it.

The subheadings for each species should be above the figures associated with or included in the legend and then referenced according to panel letter. Also including the expected band size would keep the reader from needing to flip back for reference to figure 2. For example, "panels A-E are *M. kansasii* (expected band size - 450 bp)".

As requested by the Reviewer, we have now modified the **Suppl. Fig. 1** accordingly:

1. We have divided **Suppl. Fig. 1** into **Suppl. Figs. 1-3** to present the electrophoretic gels for each mycobacterial subgroup separately.
2. We have revised the figures legends with a more detailed description.
3. We have added subfigure letters to each panel in the figures.
4. We have clarified in the legend that the alternative ladder was only applied to the panel “d”.
5. We have included subheadings above each figure panel, indicating the analyzed species, and have referenced each panel by its corresponding letter for clearer organization.
6. We have included the expected band size in the figure legend.

- Explanations of the X's in second figure panel are needed. Were these failed attempts for those shown in next panel? If so, should mention if this resolved on repeat or if multiple attempts were needed.

We have now added the lacking information. Please see **Suppl. Fig. 2**:

“Due to the image compression, some bands were either not visible or barely visible in the captured photo. To enhance the signal, a separate gel (c) was run with a double load of the selected samples.”

- Most importantly, there is no discussion or explanation of why the ~1000 bp band is missing from almost all the *M. pseudokansasii*. This needs to be addressed in the text as while the second band is different in size from any others, amplification occurred with the reference strain and in other species longer amplicons were generated making it unlikely to be length related. This needs to be discussed so if someone was implementing the assay and only got one band they would know if this is or is not a valid *M. pseudokansasii* determination.

We thank the Reviewer for pointing this out. We have now commented on this in the “**Discussion**” section. Please **lines 214-218**:

*“For some of the *M. pseudokansasii* strains (Suppl. Fig. 2), including the reference strain (Fig. 2), a ca. 1000 bp band was detected, congruently with the in silico analyses. However, this band was occasionally absent or exhibited very low intensity. Nevertheless, unsystematic lack of this band does not affect the interpretation of the assay, as the fragment of ca. 390 bp constitutes the relevant discriminating product for *M. pseudokansasii*.”*

Furthermore, we have added a disclaimer on that in the **Fig. 2** description. Please see below:

*“an occasionally occurring ca. 1000 bp band for mPCR of *M. pseudokansasii* species”.*

Minor revisions

- **Line 12**: expand to *Mycobacterium kansasii* as first presentation.
- **Line 91, 108, 115**: this section notation is not used in the paper formatting which creates confusion. Using "as described above" or "as described in ____ section".
- **Line 104**: MTBC needs to be expanded as first introduction of the abbreviation.
- **Line 134**: Add reference to **Supplemental Table 1** for the reference strains used.
- **Line 144**: add notation that K14 and K19 are isolates number 53 and 54 respectively.
- **Figure 1**: make the 590 bp product notation match between figure and text. Either state as "boxed bands" or remove the box on figure and replace with *.
- **Figure 2**: What is the * for in *M. pseudokansasii* column?

We have now introduced all these changes, and thus the manuscript has improved importantly. We thank the Reviewer for these comments.

Comments for the Author from the Reviewer #2

- **Novelty and Contribution**: The development of a one-step multiplex PCR assay is a meaningful advancement for MKC diagnostics.

We thank the Reviewer for their opinion.

Consider emphasizing how the assay compares (cost, turnaround, complexity) with existing commercial or laboratory-developed tests.

Following the Reviewer's remarks, we have now modified the "**Discussion**" section accordingly. Please see **lines 233-238**:

Contrary to existing multi-step methods, it provides fast (with a turnaround time of less than 3 hours vs. 1-3 days), affordable (with a per-reaction cost of ca. 2 EUR vs. 5-15 EUR), and reliable (98% specificity vs. 75-100%, depending on the applied method, target gene, and reference databases) identification of all MKC species (32-34). This method might be implemented in laboratories with basic molecular biology infrastructure (i.e. thermocycler, electrophoresis and gel imaging systems) as an alternative to PCR-REA and PCR-sequencing.

- **Handling of Atypical Strains:** The misidentification of the two subtype IIB isolates is important. The abstract mentions these were reassigned as *M. persicum* using a newly designed assay-however, this could confuse readers. A brief clarification that this newer assay is ancillary and not part of the main single-step method might prevent misinterpretation.

We have now clarified this in **lines 163-164**:

"[...] M. kansasii subtype IIB strains K14 and K19 (listed as 53 and 54, respectively), which were incorrectly identified as M. persicum [...]."

Re: Spectrum03267-25R1 (One-step multiplex PCR assay for identification of *Mycobacterium kansasii* complex species)

Dear Dr. Tomasz Jagielski:

Your manuscript has been accepted, and I am forwarding it to the ASM production staff for publication. Your paper will first be checked to make sure all elements meet the technical requirements. ASM staff will contact you if anything needs to be revised before copyediting and production can begin. Otherwise, you will be notified when your proofs are ready to be viewed.

Sincerely,
Natalie Whitfield
Editor
Microbiology Spectrum

Reviewer #1 (Comments for the Author):

Thank you to the authors for addressing the concerns presented. The paper is greatly improved for understanding and replication.